# Quality of Dietary Supplements Containing Plant-Derived Ingredients Reconsidered by Microbiological Approach

**DOI:** 10.3390/ijerph17186837

**Published:** 2020-09-18

**Authors:** Magdalena Ratajczak, Dorota Kaminska, Agata Światły-Błaszkiewicz, Jan Matysiak

**Affiliations:** 1Chair and Department of Genetics and Pharmaceutical Microbiology, Poznan University of Medical Sciences, Swiecickiego 4, 60-781 Poznan, Poland; dorotakaminska@ump.edu.pl; 2Department of Inorganic and Analytical Chemistry, Poznan University of Medical Sciences, Grunwaldzka 6, 60-780 Poznan, Poland; aswiatly@ump.edu.pl (A.Ś.-B.); jmatysiak@ump.edu.pl (J.M.)

**Keywords:** dietary supplements, quality, microbiological contamination, safety assessment, mycotoxins, foodomics

## Abstract

Dietary supplements cover a wide range of products, the most popular are those containing plant-based ingredients. Supplements are consumed by consumers of all ages as well as by both healthy and sick people. The lack of unified regulation in this sector increases the probability that supplements are poor chemical and microbiological quality and can be dangerous for patients. The aim of this paper is to highlight selected issues associated with the microbiological quality of dietary supplements containing plant materials. We focus on the most recent reports referring to bacterial and fungal contaminations as well as the presence of mycotoxins. Dietary supplements containing plant ingredients commonly show a variety of microbial contaminants, which might be crucial for consumer safety. They often contain microorganisms potentially pathogenic to humans. Metabolites produced by microorganisms may pose a threat to the health of consumers. Because of that, in this review, we emphasize the risk that may be associated with the lack of appropriate studies of the quality of the supplements.

## 1. Introduction

Dietary supplements (sold as food supplements, food or nutritional supplements) are meant to supplement nutritional deficiencies or to maintain a proper level of nutrients in a diet. They are the source of vitamins, minerals and other substances with nutritional or another physiological effect. They are sold in a form allowing dosing (tablets, capsules, liquids in specified doses). The European Food Safety Authority (EFSA) defines food supplements as “concentrated sources of nutrients or other substances with a nutritional or physiological effect, whose purpose is to supplement the normal diet”. The Food and Drug Administration (FDA) says the dietary supplements are products taken by mouth that contain a “dietary ingredient” intended to supplement the diet [1,2,3,4,5].

The global dietary supplements’ market is worth approximately USD 120 billion, and the annual increase during the recent years was approximately 6%. The main sales markets include Asian, North American, and European countries [3,6,7]. They are used by healthy and unhealthy persons, young and older, as well as pregnant women. Supplements can be purchased as OTC (Over The Counter Drug) products at the chemist’s, in supermarkets, at press stands or kiosks, and online [8,9]. Common, uncontrolled use of dietary supplements (including plant-based supplements) may result in health issues. Those issues may result from, for example: improper dosage, interactions with other drugs and/or supplements used, faked composition, and the existence of contamination with potentially hazardous substances or microorganisms [8].

Dietary supplements incorporating plant ingredients, are prone to be more microbiologically risky due to their origin, and treatment applied, in comparison to those obtained from bioengineering or organic synthesis. Microbiological contaminants of plant-derived supplements typically include microorganisms which occur naturally in the soil, air and water. Moreover, volatility and sensitivity of the active components of the plants to heat, UV, β and γ irradiation do not permit the use of many decontamination methods.

The safety of consumers taking dietary supplements is crucial, as part of the population includes people aged over 65, children, and persons with chronic diseases, or an impaired immune system [3,6]. The safety of consumers using dietary supplements may be interpreted from different aspects. The literature discusses issues associated with falsifying the composition of supplements, their quality, and safety (the lack of hazardous substances, and microbiological safety) [1,10].

According to the Centers for Disease Control and Prevention, 48 million persons get sick, 128,000 are hospitalized, and 3000 die from foodborne infection and illness in the United States each year. Many of these people are children, older adults, or have weakened immune systems and may not be able to fight infection normally (FDA) [11]. Dietary supplements contaminated with microorganisms may be the cause of various infections and health hazards, depending on: (1) the administration method (e.g., by mouth, on the skin, to the nose, to the respiratory tract), (2) the nature of the product (the presence of a substance which facilitates the growth of bacteria, no preservatives), (3) a group of consumers using the product (children, elderly persons, pregnant women), (4) pre-existing medical conditions and the use of different drugs [8]. Below, we present samples of infections/intoxications related to the consumption of dietary supplements documented in the literature. In 2003, in Germany, an epidemic of food poisoning of infants caused by the strain *Salmonella* enterica serotype Agona was observed. Among 39 infants demonstrating the symptoms of intoxication, 21 required admission to a hospital. The source of the pathogenic bacteria were different brands of baby teas containing such herbs as fennel, anise, or caraway. In the course of epidemic investigation, 18 different brands of herbal teas for babies were tested. *S*. Agona was identified in 8% of samples, most of the samples contained anise. Those were the products of different manufacturers; however, all of them imported anise from the same supplier. The contamination of anise with bacteria was caused by fertilizing the crops with manure [8].

Another example of infections associated with the use of supplements contaminated with *Salmonella* bacteria type, are the cases of approximately 200 infected persons noted by FDA in 2018, out of which 38% had to be admitted to hospital. The source of *Salmonella* bacteria was Mitragyna speciosa-kratom, a plant naturally grown in Thailand, Malaysia, Indonesia, and Papua New Guinea, and used in natural medicine. The presence of bacteria was confirmed in 33 out of 66 samples taken from distributors and retail shops [12,13].

The presence of *Salmonella* bacteria is confirmed in different dietary supplements containing plant ingredients. In January 2018, due to *Salmonella* bacteria two different products were recalled (available online). One of them, intended for men, contained: psyllium husk, flax seeds and chia seeds, whereas the other product used for arthritis, contained a number of plant ingredients: Turmeric (relieve arthritis pain and inflammation), Boswellia (a tree gum derivative), Ashwagandha (an herb used in Ayurvedic medicine to treat arthritis), Yucca (which is said to have anti-inflammatory effects).

Dietary supplements contamination with fungi may be particularly dangerous for patients in risk groups (after a transplant, in immunosuppression, with neoplasms, with reduced immunity, and premature infants). A development of invasive mycosis after consuming products contaminated with fungi, is observed more and more frequently [14]. Fungi of the types *Aspergillus* and *Candida*, which most frequently infect the respiratory system, alimentary system, and blood, are the most frequent causes of invasive fungal infections in persons with reduced immunity. Infections with mould fungi such as *Mucor, Rhizopus, Absidia* may result in considerable morbidity and mortality [14].

In 2014, in a hospital in Connecticut, a fatal case of stomach-intestine mucormycosis in a premature infant born in 29th gestational week was reported. The infant was administered a dietary supplement to prevent necrotizing enterocolotis. Tests of the supplement sample indicated that the product was contaminated with mould fungi of the type *Rhizopus*, able to cause mucormycosis [15]. In literature, we can also find a description of a case of a bone marrow transplant receiver in which liver mucormycosis was developed as a result of taking many naturopathic, oral supplements. Fungi, including *Aspergillus*, *Rhizopus* and *Mucor*, were identified in 4 out of 10 different supplements consumed by the patient. *Mucor indicus* strains aspired from the patient’s liver and the dietary supplement were genetically identical [16]. Other authors describe a case of stomach and intestine absidiomycosis in a 10-year old girl with leukaemia as a result of the use of a probiotic supplement. The patient was taking the supplement to restore proper intestinal microbiota. In the taken preparation, a high concentration of fungi *Absidia corymbifera* and the presence of *Aspergillus flavus* and *Candida* spp. was found. Genetic tests highly suggest that the probiotic capsules were the source of infection in the child [17]. The composition of probiotic products should include strictly characterized strains of microorganisms of proven clinical effectiveness. In order to reliably confirm the identity of probiotic strains at the species level, tests of such preparations should include not only phenotypic but also genotypic methods.

The above examples confirm that the presence of microorganisms in dietary supplements may be the cause of serious diseases. Therefore, the aim of this paper is to analyze selected issues associated with the microbiological quality and safety of use of dietary supplements containing plant materials. We present the most recent reports referring to bacterial and fungi contaminations and the presence of mycotoxins.

## 2. Legal Aspects

The general accessibility and wide-scale advertising campaigns, cause customers’ increased interest in supplements. There are no uniform regulations defining which products can be classified as dietary supplements, food products, or drugs. In the European Union, food supplements are classified as foodstuffs and thus all food law applies to food supplements. Since 2002, the European Union has enforced a legal and regulatory framework for these products with the Food Supplements Directive 2002/46/EC. The directive defines a food supplement as “foodstuffs the purpose of which is to supplement the normal diet and which are concentrated sources of nutrients or other substances with a nutritional or physiological effect, alone or in combination, marketed in dose form, namely forms such as capsules, pastilles, tablets, pills and other similar forms, sachets of powder, ampoules of liquids, drop dispensing bottles, and other similar forms of liquids and powders designed to be taken in measured small unit quantities”. Directive contains a list of nutrients and their chemical forms able to be used in food supplements. The maximum levels and conditions of use for other substances, such as botanicals, botanical preparations and bioactive substances, are not specified in it [5].

In the United States, good manufacturing practices within manufacturing, packing, labelling, storage are applied. Before supplements are introduced for trading, they are not required to be approved by Food and Drug Administration (FDA) [18]. However, the manufacturers and distributors of dietary supplements are responsible for the safety of their products before introducing them to trade. FDA may take actions to recall products, but the law only permits it if it has been determined that a product is hazardous, falsified, or poorly labelled.

The problem of diet supplement falsification in the European Union is discussed by Kowalska et al. [19]. Based on the analysed data, the authors claim that most of the reported inconsistencies are associated with improper labelling, in particular with falsifying the composition, nutrition, and health claims. The frequency of reported cases of inconsistencies in food supplements, as compared with other food products, is high. Frauds associated with the falsification of dietary supplements pose a threat to public health. According to the applicable law, a dietary supplement is not a medicinal product. However, it is sold in the form of tablets, capsules, sachets, so it has the same form as drugs. In combination with unfair, misleading advertisement, it may give people with little or no knowledge about the product, a wrong idea of its healing properties.

## 3. Microbiological Quality of Dietary Supplements Containing Plant Materials

Herbal supplements may contain ingredients made of different parts of a plant, or herbal extracts. According to the World Health Organisation (WHO), a medicinal plant is a plant, in which at least one part (stalk, root, fruit, flower) may be used for medical purposes [9]. Herbal supplements are regulated by the food legislation and they are not subject to clinical trials to check their effectiveness or toxicity. Microorganisms present in a product may impair its stability, cause a change of its physical properties, or inactivate the active ingredients. Moreover, microorganisms may decompose supplement ingredients to toxic products.

### 3.1. Microbiological Contamination Sources

Herbal preparations may be contaminated with a wide spectrum of microorganisms [2,20,21,22,23,24]. The microorganisms present in them may be classified as primary (forming the plant’s microbiota), and secondary contaminations, i.e., such contaminations which have penetrated the product during its widely understood manufacturing. Plant-associated microbiota are complex. The microorganisms occupying above-ground (phyllosphere), below-ground (rhizosphere) and internal (endosphere) parts of plants are very different.

Plants are also contaminated with microorganisms naturally present in soil, air, and water. Microbiological contaminations include vegetative bacteria and their endospores, yeasts and moulds, and their spores, viruses and protozoa. [4,6,8,11].

Plant storage is an important part of the manufacturing process because external environmental factors may affect their physical, chemical and biological properties. Storage in a poorly ventilated warehouse may cause, e.g., air temperature and humidity to rise; dried plant materials easily absorb moist, thus making the plants more susceptible to the development of moulds and production of mycotoxins. Mould fungi are predominant contaminations, naturally present on plants, multiplying after collection, if the relative humidity is not controlled during storage [24,25]. In the study [25], dried *Plantago lanceolata* leaves were subjected to atmospheres of different relative humidity (75, 45 and 0%) for 24 weeks. It was observed that higher humidity was conducive to the development of mold on the leaves. A total of 32 samples of available black and green teas were analysed in a study in Italy. The quantitative studies showed that in over 80% of samples, there were noted microorganisms presence in the amount from 1.0 × 10^2^ to 2.8 × 10^5^ CFU/g. Most of the identified microorganisms were classified as the family *Bacillaceae*. The qualitative analysis allowed identification of the following strains: *Pseudomonas psychrotolerans*, *Staphylococcus warneri*, *Pantoea gaviniae* and one strain *Clostridium perfringens*, whose ability to produce toxins may cause harmful consequences for the consumers [4]. The World Health Organisation (WHO) recommends that fresh materials of medicinal plants be stored in appropriately low temperatures, preferably in the temperature between 2–8 °C, frozen products should be stored in temperatures below −20 °C. Therefore, plant growers must be trained within good manufacturing practice (GMP), good agricultural and collection practice (GACP), and proper storage [8,21].

Microorganisms may be responsible for biodegradation of plant materials, reduce the products’ validity period and contribute to financial loss; therefore, the following rules must be kept in mind: (1) collection of plants under unfavorable weather conditions (wet and cold weather) and limitation of contact with soil must be avoided; (2) proper drying, storing and packing procedures must be observed; and (3) the potential risk sources must be assessed [9,20,24,26,27].

In the manufacturing of herbal medicines, it is essential to assure their quality. To avoid microbial contamination, a high level of sanitation and hygiene during manufacture is necessary. In order to inactivate or remove any objectionable contaminant possibly present, pre-manufacturing process such as drying, extraction, heat treatment, irradiation, or gaseous sterilization treatment are required. Drying is a process in which the moisture content of the plant is reduced which helps in preventing the enzymatic and microbial activity and finally ensures a good shelf life of the product. It is an appropriate method for the preservation herbal plants. In addition, it contributes to reduce the weight and volume of the plant with positive consequences for transport and storage. The choice of drying conditions depend on the moisture content of tissue at harvest, the plant parts used, and the temperature best suited for preservation of the requested ingredients. For this reason, adequate dryers are needed, using temperature, velocity and humidity values for drying air that provides a rapid reduction in the moisture content without affecting the quality of the active ingredients of medicinal plants. Medicinal plants can be dried in a number of ways: in the open air (shaded from direct sunlight); placed in thin layers on drying frames, wire-screened rooms or buildings. High hydrostatic pressure (HHP) processing, is a method where the material is subjected to elevated pressures applied. HPP is a technology which improves the microbial safety and shelf life of the product. It is quite commonly applied to the food processing industries. This is a non-thermal technology that leads the destruction of undesirable micro-organisms and inactivates enzymes without affecting the bioactive constituents. Gram-positive bacteria are highly resistance to high pressure as compared to the gram negative. Similarly, microbial cells in the stationary growth phase are highly resistant to pressure as compared to the exponential phase. In recent years, a new technology, instant controlled pressure drop (DIC for Détente Instantanée Contrôlée) has been developed as a decontamination process, particularly for heat-sensitive solids and powders. In this method, the material is treated for a shorter duration leading to the instantaneous drop of pressure to vacuum leading the evaporation of moisture and exploding of the microorganisms in both spores and vegetative forms. The removal of moisture and microorganisms leads to improved quality and shelf life of the herbal medicinal product [20,26,27,28]

Contamination of end products may also result from their improper handling during manufacturing, packing, and transportation. The hypothetical sources of bacterial contaminations may be cross-contaminations, i.e., microorganisms coming from processing equipment, from such materials as plastics, glass and other, who come in contact with the manufactured product. The main contaminants are aerobic, mesophilic microorganisms: intestinal bacteria, yeasts, and moulds.

### 3.2. Quantitative and Qualitative Bacterial Contamination of Dietary Supplements

Microbiological purity tests should include quantitative tests (the estimation of the total count of aerobic bacteria and fungi), and qualitative tests (detecting the presence of indicator potentially pathogenic microorganisms). The presence of microorganisms in dietary supplements was confirmed by many researchers [2,6,7,8,21,22,23,24].

In their research, the authors of this paper have analysed 122 samples of dietary supplements containing plant ingredients, randomly purchased at Polish chemists’. The preparations contained: blueberry fruit (n = 50), banana powder (n = 10), midland hawthorn fruit (n = 10), Jerusalem artichoke root (n = 14), red raspberry fruit (n = 10), linen seed (n = 20), globe artichoke leaves extract (n = 8). The samples tested showed high content of aerobes, with their count ranging from 1.0 × 10^1^ CFU/g to 6.0 × 10^6^ CFU/g. The highest count of aerobic microorganisms per one gram of preparation was present in the samples containing: linen seeds, artichoke, and blueberry fruit. They were mainly the genera of *Bacillus*, *Staphylococcus*, *Micrococcus* and *Enterobacteriaceae* family. Five samples contained Gram-negative bacteria in the amount >10^3^ CFU/g. The most frequently isolated strains included: *Enterobacter cloacae, Klebsiella pneumoniae*, *Klebsiella oxytoca*. In two samples, the presence of *E. coli* was observed [2]. The presence of bacteria from *Enterobacteriaceae* family in the tested supplements is mainly associated with human or animal feces used as plant manure.

In other study, 29 herbal products purchased from local shops in the USA were tested (including those containing: ginger n = 5, rosemary (*Rosmarinus officinalis*, n = 2), garlic (*Allium sativum*, n = 7), turmeric (*Curcuma longa*, n = 4), onion (*Allium sepa*, n = 5), mustard (*Brassica hirta*, n = 4), goldenseal (*Hydrastiscan-adensis*, n = 1) and (*Brassica juncea*, n = 1). Twenty-eight products were contaminated with bacteria which in the supplements containing garlic, onion, and turmeric exceeded the count of 10^4^ CFU/g. The most frequent contaminants were *Bacillus* spp. Two products (with garlic and onion) contained intestinal bacteria: *Stenotrophomonas maltophilia* and *Enterobacter cloace*. Samples containing rosemary were contaminated with *Erwinia* spp. Samples with mustard contained *S. aureus*, whereas mustard seeds contained other species of *Staphylococcus* [24]. Tests of over 150 different herbal preparations obtained from retail sales points in Nigeria showed that approximately 3% of samples were contaminated with aerobic bacteria in the range between 1.8 × 10^8^ and 2.25 × 10^8^ CFU/g, whereas in nearly 13% of samples, the bacteria count was approximately 10^7^ CFU/g. The most frequently occurring pathogenic bacteria included *S. aureus*, *E. coli* and *Salmonella* spp. [29]. Among 132 herbal products tested in Brazil, more than half of the samples (51.5%) contained bacterial contaminations. The most frequently isolated microorganisms were *S. aureus* (49.2%), *Salmonella* spp. (34.8%), *E. coli* (25.8%) and *P. aeruginosa* (14.4%) [6]. Additionally, Czech et al., who tested 138 samples of herbal drugs and discovered the presence of pathogenic bacteria (*E. coli* in 4 samples and *Campylobacter jejuni* in 2 samples) [23]. Table 1 shows bacterial contamination of dietary supplements.

Bacteria spores were identified in dietary supplements by different authors [2,8,24,30,31,32]. Martins et al. confirmed the presence of *B. cereus* spores in 96.8% of samples, and in 19.2% of the samples, the count exceeded 10^3^ spores per gram. The highest levels were revealed in corn silk samples (>10^7^ spores/gram) [32]. Bianco et al. have proven that out of 200 samples of camomile tea, 7.5% were contaminated with *C. botulinum* spores [31]. Other results were obtained from tests of preparations containing camomile. It was proven that 8 out of 13 (61.5%) samples of camomile contained *C. botulinum* spores [8]. The authors concluded that consumption of camomile tea may be associated with a risk of intoxication in infants and toddlers. Martins et al. have proven that *C. perfringens* spores were present in 83.9% of tested herb samples. Linden flowers and orange tree leaves were also contaminated with *C. perfringens* spores [32].

In the studies cited above, researchers report a different percentage of microbiologically contaminated samples (from about 50 to 100%). In quantitative studies, they find different levels of contamination, and qualitative tests confirm the presence of both non-pathogenic and pathogenic microorganisms. Microbiological tests often reveal the presence of bacteria of the genera *Bacillus* and *Clostridium*, which is related to their common occurrence in the environment. The presence of these bacteria is associated with the production of endospores resistant to unfavorable physical conditions, such as high temperature and low humidity. Therefore, they can survive long periods in a hibernated condition on a product. Most species of the genus *Bacillus* are saprophytic bacteria but *Bacillus cereus*, often isolated from plants, is associated with foodborne illnesses. Likewise, most bacteria of the genus *Clostridium* are saprophytes. As commensals, they are part of the physiological intestinal flora of humans and animals. Few species of this genus are human pathogens (e.g., *C. botulinum*, *C. perfringens* may cause food poisoning). The presence of *Enterobacteriaceae* family bacteria, mainly *E. coli*, *K. pneumoniae, Enterobacter* spp. and *Salmonella* spp. indicates a contamination of faecal origin (human or animal), and may be an indicator of improper hygienic conditions at manufacturing site. It may also indicate the use of natural fertilizers for farming. *S. aureus* is not a natural contamination of plant raw materials; its presence may be associated with human vectoring.

Water, poor worker hygiene, and sanitation practices during harvesting, sorting, processing, packaging and transportation can also be sources of contamination.

### 3.3. Dietary Supplements Contamination with Fungi

Several problems arise in the case of fungi contamination. Mould fungi are widely spread in the natural environment and therefore are often present in raw materials of plant origin. The spores produced by mould fungi are easily airborne and they are not always removed in technological processes used for supplements manufacturing. Storage and transport conditions of the plant materials affect the occurrence of fungal contamination in the end product.

It is known that approximately 300 out of 1.5 million fungi present on Earth are hazardous to human health and may cause a variety of health conditions, from allergic reactions to life-threatening invasive infections [27]. Many fungi synthesise secondary metabolites—mycotoxins, some of which may cause, for example, acute poisoning, liver diseases, neural tube defects, neoplasms. Fungi which contaminate or deteriorate food mainly include *Alternaria*, *Aspergillus*, *Candida*, *Fusarium* and *Mucormycetes* [38,39,40]. Studies carried out by the authors of this paper indicate that significant percentage (86.8%) of samples of dietary supplements with plant ingredients tested (n = 122) were contaminated with fungi. The most frequently isolated mould fungi included the types: *Aspergillus* spp. (81.8%), *Alternaria* spp. (47.0%), *Penicillium* spp. (28.8%), *Fusarium* spp. (13.6%), *Cladosporium* spp. (6.0%) and other (yeasts, *Rhizopus* spp., *Mucor* spp.) (15.9%). One hundred percent of supplements containing blueberry fruit, banana powder, midland hawthorn fruit, red raspberry fruit, and artichoke leaves extract were contaminated with fungi. However, only in five samples the contamination level exceeded 10^4^ CFU/g [2].

The problem of microbial contamination of dietary supplements with moulds was discussed by Tournas et al. They reported that seventy eight percent of the ginseng herb supplements, 100% of the Siberian, 56% of the Chinese and 48% of the American ginseng root samples showed fungal contamination. Fungi found in the ginseng herb were *Alternaria alternate*, *Aspergillus niger*, *Aspergillus* spp., *Cladosporium* spp., *Penicillium* spp., *Rhizopus* spp. and yeasts [36]. In another paper, the same authors tested 138 different dietary supplements containing such plant ingredients as medicago sativa, Circu-Care, coriander, cumin, echinacea, garlic, ginger, ginko, juniper berries, and valerian. The highest level of contamination with fungi was discovered in supplements containing medicago sativa (5.6 × 10^6^ CFU/g), and the lowest in ginger supplements (1.0 × 10^2^ CFU/g). Research carried out by Tournas have shown that the most frequently isolated fungi were *Aspergillus*, *Eurotium* and *Penicillium*. More seldom *Alternariaalternata*, *Fusarium*, *Cladosporium*, *Rhizopus* spp. and *Phoma* spp. [22].

In Brazil, 91 samples of medicinal herbs were tested, and it was concluded that half of them were contaminated with moulds. The prevailing genera were *Aspergillus* and *Penicillium*, which constituted 89.9% of all identified moulds. Those fungi are extremely important due to the production of mycotoxins [41]. Similarly, in the study carried out by Bugno et al. [42], it was found that 50% of 91 tested medical herbs were contaminated with fungi. The researchers have proven, that the plant parts that were most frequently contaminated with mould fungi were the aerial parts (leaves, flowers). Whereas Croatian researchers Cvetnić and Pepelnjak [25], studied the underground parts of herbal plants (53 samples) and have discovered that their roots and rhizomes are also contaminated with moulds. Other authors, after testing 132 samples, observed the growth of mould fungi in 35.6% of herbal supplements, and in 31.8% of samples, fungi count was higher than 10^5^ CFU/g [8].

Contrary to moulds which are strong allergens and developers of mycotoxins, little is known about the presence of yeasts in medical herbs, dietary supplements and other herbal products. In a study of medical herbs carried out in Portugal, contamination with yeast-like fungi was found. Martins et al. [32] isolated *Rhodothorula glutinis* from chamomile (*Matricaria chamomilla* L.) (7.7% of samples), flower of linden (*Tillia grandifollia* L.) (15.4%) and penny-royal mint (*Mentha pulegium*) (100.0%). The authors have also grown *R. mucilaginosa* which was present in camomile and orange leaves. Such species as *Cryptoccocus laurentii* and *Cryptococcus albidus* were identified in chamomile, and *Candida guilliermondii* in Kloeckera japonica [6,8]. The presence of yeast-like fungi was noted/revealed also in the study carried out by Halt and Klepec [43]. *Saccharomyces* and *Candida* fungi were isolated the most frequently.

Moulds are capable of surviving in environments of very low humidity. Moreover, they can multiply in the presence of a very limited amount of nutrients. Multiplying moulds can not only reduce the quality and usefulness of dietary supplements. The presence of fungi could also lead to the secretion of toxic secondary metabolites.

## 4. Mycotoxins

Fungi of the types: *Aspergillus*, *Penicillium*, *Fusarium* and *Alternaria*, under favorable conditions, are capable of producing secondary metabolites—mycotoxins [7,42,44,45,46,47,48].

Over 40 different mycotoxins were identified in herbal preparations. Effective control of contamination with mycotoxins requires adequately sensitive and precise analytical methods. An excellent overview of analytical methods applied for determining mycotoxins is presented in the paper by Zhang et al. [7]. One fungus species can produce many different mycotoxins, and the same mycotoxin can be produced by several different species. Contamination with mycotoxins produced by fungi can take place at different phases of the manufacturing process, starting with the plants growing on a field, through the collection, transport, and finally, during the storage and drying phase [49,50,51]. The presence of mycotoxins may (1) affect the effectiveness of active substances; (2) lead to dangerous interactions; (3) intensify adverse effects; (4) cause mycotoxicoses.

Aflatoxins (AF), ochratoxin A (OTA), zearalenone (ZEN), trichothecenes and fumonisins are specifically important due to their frequent presence in dietary supplements, and proven adverse effect on human health [46,52,53]. Such mycotoxins demonstrate carcinogenic, teratogenic and mutagenic activity. They can have an adverse effect on urinary system, reproductive system, immune system, and central nervous system [44,54]. Aflatoxin AFB1 is the strongest carcinogenic factor existing in nature. The International Agency for Research on Cancer (IARC) declared that aflatoxins: AFB1, AFB2, AFG1 and AFG2 demonstrate explicit, proven carcinogenicity for human, and classified it to group 1 of carcinogenic factors [55,56]. Ochratoxin A causes oxidative DNA destruction leading to mutagenesis and potential carcinogenesis [57,58,59]. Therefore, is classified by IARC as a potential carcinogenic factor for humans [56].

Fumonisins are produced by fungi of the type *Fusarium* (*Fusarium proliferatum* and *Fusarium verticillioides*). Among 28 compounds classified as fumonisins, the most frequent is fumonisin B1 (FB1) [60]. Toxins FB1 and FB2 are listed as potential carcinogenic substances in group 2B of IARC Classification [56]. Moreover, trichothecenes (HT-2, T-2) may cause skin irritation, stomach and intestine problems (nausea, contractions, vomit), disturbed mitochondria functions and hypothrophia of spleen and thymus [44]. It was also proven in vitro that those toxins may cross the blood–brain barrier, which may result in neurotoxical activity [61]. While zearalenone (ZEN) shows a structural resemblance to human sex hormone 17β-estradiol, influences oestrogen receptors and causes reproductive issues, both in human and in farm animals [44]. IARC found limited evidence of ZEN carcinogenesis in animal models, classifying them in group 3 [56].

The reports on the presence of mycotoxins in dietary supplements differ by the percentage of contaminated samples, and the revealed level. Mycotoxins are found in preparations based on different medicinal plants [49,62,63,64,65,66,67,68,69] (Table 2).

Veprikova et al. tested 69 samples of herbal supplements (containing, among other milk thistle, red clover, linen seed, soy, barley grass, nettle, goji berries) for the presence of 57 different mycotoxins. The authors have concluded that most of them 96% (66 out of 69) contained at least one mycotoxin [64]. In many cases, a coexistence of different toxins was observed. Santos et al. also observed a coexistence of different mycotoxins in the 84 samples of dietary supplements tested [74]. Among the preparations tested (containing milk thistle, camomile, valerian, senna, rhubarb, ginkgo biloba), 99% were contaminated with toxin T-2, 98% with zearalenone (ZEN), 96% with afllatoxin, 63% with ochratoxin A, 62% with deoxynivalenol, 61% with citrinin and 13% with fumonisins (FB). All samples tested contained more than one mycotoxin [74]. Such coexistence of many different mycotoxins may be particularly dangerous because we do not know their interactions. The effect may be additive, or even synergic [64].

Veprikowa et al. found the highest concentration of mycotoxins in supplements based on milk thistle. In all 32 samples, they have noted the presence of at least one mycotoxin. The most frequently found mycotoxins included DON, ZEN, HT-2 and T-2. In approximately 58% of samples, the coexistence of more than 12 mycotoxins was confirmed [64]. Other authors also tested mycotoxins content in supplements based on milk thistle [57,58]. However, the percentage values of samples containing mycotoxins were lower. Tournas et al. report that 15 out of 83 tested samples of supplements with milk thistle contained aflatoxins in the amounts of 0.04 to 2 μg/kg [62]. Whereas Arroyo-Manzanares et al. have analysed seven samples of dietary supplements based on milk thistle for the presence of 15 different mycotoxins. In two of the tested samples, they have found HT-2 and T-2 toxins. T-2 amounts were 363.0 and 453.9 μg/kg, and HT-2 822.9 and 933.7 μg/kg [63].

Solfrizzo et al. have analysed 24 dietary supplements containing grape pomaces for contamination with ochratoxin A. The presence of OTA at the level <1.16–20.23 μg/kg was confirmed in 75% of tested samples [68]. Other authors also confirm the presence of ochratoxins in the supplements tested [42,53,61,63]. In contrast to the results presented above, many reports and analyses indicate that only a small portion of the analysed samples is contaminated with toxins [49,72,81].

The presence of mycotoxins in the consumed dietary supplements not necessarily poses a threat to the consumer. The levels found may fall under the allowed limits, and may not be seriously hazardous to human health. Different studies confirmed that the dietary supplements’ share in the total weekly consumption of toxins is in most cases low [49,69]. However, Veprikowa et al. have come to contrary conclusions when analysing the mycotoxins contamination level in preparations based on milk thistle (used to support the work of liver); she has concluded that by using recommended doses of such preparations one can consume as much as 75% of the daily—tolerable intake (TDI is 0.06 µg/kg bw) of HT-2 and T-2 [64].

Frequent findings of several mycotoxins presence at the same time in different concentrations (where the toxic effect of the coexistence is unknown) indicate the need to reinforce the quality control of such products [64,68,74].

## 5. Foodomics Technologies for Mycotoxins and Microorganisms Detection

To date, methods based on microbial culture and biochemical identification (membrane filtration, plate-count methods, most-probable-number method) are recommended for the quantitative and qualitative evaluation of microorganisms in dietary supplements. However, methods based on molecular biology and omics technologies allow for more sensitive and specific food contaminants identification. Foodomics has been first defined by Cifeuntes as “discipline that studies the Food and Nutrition domains through the application of omics technologies” [83]. This new branch of science should provide total knowledge about food’s functionality, origin, nutritional value, safety and quality. Recently, there can be observed huge progress in sophisticated analytical tools, which are used in genomics, metabolomics and proteomics studies. Therefore, the great opportunity to analyze the quality and safety of food of animal as well as plant origin, has appeared. Deep analysis of genome, proteome, metabolome can provide reliable information about contamination with pathogenic microorganisms, their metabolites or mycotoxins.

Mycotoxins mainly can be classified as metabolites. Therefore, according to these contaminants, foodomics studies are mostly based on metabolomics technologies. Gas chromatography coupled to mass spectrometry (GC–MS) is the method of choice to analyse volatile metabolites, while liquid chromatography coupled to mass spectrometry (LC–MS) is required to characterize non-volatile metabolites. Since all currently known mycotoxins are non-volatile, only few papers described the use GC–MS for their analysis [84,85,86]. Recently, two reviews summarized the use of high-resolution MS (HRMS) in the topic of mycotoxins (such as Orbitrap technology [87] and ion mobility MS [88]). Quantification analysis of mycotoxins require targeted metabolomics. Regarding low levels of mycotoxins (often at sub ppb levels), methods need to be characterized by proper sensitivity and specificity. For this purpose, LC–MS platforms usually are based on triple quadrupole mass detectors (QqQ), HRMS, like time-of-flight (TOF), or Orbitrap. The analytical methods are usually developed to achieve limits of quantification (LOQs) and detection (LODs) in agreement with regulatory authorities for the official control methods [88]. However, these parameters are strongly dependent on the type of used instruments and their actual condition. The application of LC-QqQ-MS enabled to identify 295 different bacterial metabolites and mycotoxins in different food matrices: apple puree, hazelnuts, maize and green pepper [89]. LOD was different for all detected analytes and dependent on the used matrix. For example, LOD for chanoclavine was 0.04 μg/kg for hazelnuts and 0.1 μg/kg for apple puree. While LOD for patulin was 1.2 μg/kg for hazelnuts and 35.9 μg/kg for apple puree. The proficiency tests materials were performed, and they allowed for example to detect 1406 μg/kg of fumosin B or 7 μg/kg of ochratoxin A in maize. The paper presents detailed information with LODs and LOQs values as well as levels of detected analytes in tests involving different matrices. Another analytical instrument Orbitrap was used for instance to quantify 20 mycotoxins and screen 200 fungal metabolites in food including maize and wheat [90]. LC-TOF-HRMS proved to be robust method for determination levels of 26 toxins on 147 samples of the grain of cereals [91].

Non-targeted metabolomics allow for identification of unknown mycotoxins. During such analysis, even thousands of different features might be observed. Therefore, non-targeted experiments are usually conducted using HRMS such as TOF-MS or Orbitrap. TOF-HRMS was used for identification, so far unknown, mycotoxins of *Aspergillus flavus* [92]. In the literature, there are examples of using Orbitrap for identification of mycotoxins in green tea and royal jelly supplements [72], bakery products [93] or wheat [94]. LODs for analytes identified in the bakery products were below 6 μg/kg and LOQs were in the range 5–100 μg/kg [93]. Detailed comparison between metabolomics technologies for mycotoxins analysis are presented in the review by Rychlik et al. [84].

Multi-omics approaches have become useful method for pathogens contaminants analysis. They can provide crucial information about activities of pathogens and their metabolites. Moreover, some relevant proteins and metabolites might be helpful in trace microbial infection and allow for functional analysis. Importantly, by the employment of the most sophisticated achievements of omics technologies, it is possible to identify and scan post translational modifications and low abundant proteins, peptides and metabolites, which might be crucial for monitoring of food and dietary supplement safety [94,95,96,97]. Genomics studies are mainly based on DNA microarray technology [98]. Whereas, during proteomics and metabolomics studies GC–MS [99], LC–MS [100] and HRMS [101] are usually used. Omics technologies might be used for investigations of model bacteria/fungi under stress conditions. Examination of adaptation and reaction for stress conditions leads to identification of cellular and metabolic biomarkers. Correlation between such biomarkers and adaptive stress allow to predict the influence of environmental and food-related changes on the resistance and survival abilities of microorganisms [102]. Moreover, molecular biomarkers are useful indicators for early detection of food pathogens [103].

In the literature, there can be found many examples of using omics technologies to identify contaminants in food. The metabolomics approach was for instance developed to detect *E. coli* O157:H7, *Salmonella* Hartford, *Salmonella* Typhimurium, and *Salmonella* Muenchen in chicken and ground beef [99]. Protein analysis allowed for identification of *Aeromonas hydrophila* and *Yersinia enterocolitica* in fresh vegetable salads [104]. While genomics tools were presented in the recent review as reliable techniques for detection of foodborne pathogens [98].

Specific and sensitive methods for detection and monitoring dangerous contaminants such as mycotoxins and microorganisms are crucial to ensure dietary supplements safety. Therefore, during recent years, the improvement of novel omics approaches such as genomics, proteomics or metabolomics, resulted in development of another omics branch of science—foodomics. Since, the identification of specific biomarkers—proteins or small molecules, produced by unknown bacteria or fungi, maybe sophisticated way to prevent foodborne diseases, in the further use of foodomics may be essential to verify food and dietary supplements contaminations.

## 6. Summary

This paper contains a current overview of literature concerning the safety of use of dietary supplements containing plant ingredients. We have paid special attention to microbiological contaminations and their consequences for patients. The reports on the presence of microbiological contamination in dietary supplements containing plant-derived ingredients differ by the percentage of contaminated samples, and the revealed level. The presence of microorganisms (such as *E. coli*, *Enterobacter* spp., *Salmonella* spp. and *S. aureus*), can be potentially harmful to human health. Botanicals may be microbiologically contaminated during cultivation, harvesting, packing and distribution. Therefore, it is important that microbiological contamination be minimized throughout the manufacture of dietary supplements containing plant materials.

All countries should use reasonable efforts to make the consumers trust the manufacturers of dietary supplements, and be sure that what they get is a safe product of the highest quality and the main goal of dietary supplement manufacturers should be to ensure consumer safety. The aim of quality control is to ensure that each single product, no matter where it is manufactured and purchased, is free from contaminations and meets all quality standards. As herbal medicinal products are complex mixtures originating from different sources, enormous efforts of many entities are required to ensure fixed and appropriate quality at each phase of the manufacturing process. With a normalised manufacturing process, the quality of dietary supplements around the world will be maintained at the highest level, and as a consequence, patients will receive products safe for their health.

## Figures and Tables

**Table 1 ijerph-17-06837-t001:** Qualitative and quantitative microbiological contamination of dietary supplements.

Plant Component	Bacterial Contamination		Fungal Pollution		Ref.
	Quantitative TAMC (CFU/g)	Qualitative	Quantitative TYMC (CFU/g)	Qualitative	
Lucerne (alfalfa) leaves	5.2 × 10^6^–3.8 × 10^7^	Aerobic plate counts	4.4 × 10^5^–5.6 × 10^6^	*Cladosporium* spp., *Fusarum* spp., *Aspergillus flavus*, *Aspergillus niger*, *Penicillium* spp., Yeasts	[22]
Ginger root	<10^2^–1.0 × 10^2^	Aerobic plate counts	1.5 × 10^2^–5.4 × 10^5^	*Aspergillus niger*	[22]
Ginkgo	<10^2^–3.2 × 10^3^	Aerobic plate counts	<10^2^–3.8 × 10^5^	*Aspergillus* spp., *Eurotium chevalieri*, Yeasts	[22]
Echinacea herb	<10^2^–2.4 × 10^3^	Aerobic plate counts	<10^2^–4.6 × 10^5^	*Alternaria alternate*, *Fusarum* spp., *Aspergillus* spp., *Aspergillus niger*, Yeasts	[22]
	1.9 × 10^6^		8.2 × 10^3^		[18,23]
European blueberry fruit	<1.0 × 10^1^–2.0 × 10^5^	*Bacillus* spp., *Micrococcus* spp., *Staphylococcus* spp., *Enterobacteriaceae*	1.0 × 10^1^–7.0 × 10^4^	*Alternaria* spp., *Fusarum* spp., *Aspergillus* spp., *Cladosporium* spp., *Penicillium* spp.	[2]
Raspberry fruit	<1.0 × 10^1^–3.0 × 10^2^	*Bacillus* spp., *Micrococcus* spp., *Staphylococcus* spp.	1.0 × 10^1^–4.0 × 10^4^	*Alternaria* spp., *Fusarum* spp., *Aspergillus* spp., *Penicillium* spp.	[2]
Jerusalem artichoke root	5.0 × 10^1^–7.0 × 10^5^	*Bacillus* spp., *Micrococcus* spp., *Staphylococcus* spp., *Enterobacteriaceae*	<1.0 × 10^1^–7.0 × 10^2^	*Alternaria* spp., *Fusarum* spp., *Aspergillus* spp.	[2]
Aristolochia repens	5.4 × 10^5^	*Citrobacter* spp., *Klebsiella aerogenes*, *Bacillus subtilis*	3.1 × 10^6^	*Aspergillus fumigatus*, *Absidia* spp.	[33]
Angylocalyx oligophyllus	3.5 × 10^6^	*Bacillus subtilis*, *Citrobacter* spp., *Staphylococcus epidermidis*	7.5 × 105 ± 0.03	*Mucor* spp.	[33]
Zingiber officinale	2.0 × 10^6^	*Acinetobacter* spp., *Pseudomonas aeruginosa*, *Bacillus subtilis*	Nil	-	[33]
	1.0 × 10^3^	*Bacillus spp., Staphylococcus* spp.	2.3 × 10^2^	*Aspergillus* spp.	[34]
Securinega virosa	4.3 × 10^5^	*Bacillus subtilis*, *E. coli*	7.1 × 10^5^	*Mucor* spp., *Penicillium* spp.	[33]
Nesogordonia papaverifera	6.3 × 10^6^	*Pseudomonas aeruginosa*, *Citrobacter* spp.	7.1 × 10^6^	*Aspergillus niger*, *Mucor* spp.	[33]
		*Bacillus* spp., *Staphylococcus epidermidis*			
Astralagus savcocolla	1.2 × 10^6^	*Bacillus* spp., *Staphylococcus* spp.	2.1 × 10^4^	*Aspergillus fumigatus*, *Aspergillus flavus*	[34]
Matricavia chamomiia	1.0 × 10^5^	*Enterobacter cloace*, *Bacillus* spp.	1.7 × 10^3^	*Aspergillus flavus*	[34]
Calligonum comosum	3.7 × 10^2^	*Bacillus cereus*	1.0 × 10^5^	*Aspergillus flavus*	[34]
Matricaria chamomilia	4.0 × 10^5^	*Clostridium botulinum*	1.7 × 10^3^	*Aspergillus flavus*	[34]
	1.7 × 10^6^		2.5 × 10^3^	Yeasts	[23]
	3.5 × 105	*E. coli, Bacillus* spp., *Micrococcus* spp.	-	*-*	[35]
American ginseng root	<10^2^–4.5 × 10^4^	*Bacillus* spp.	<10^2^–4.3 × 10^5^	*Penicillium* spp., *Rhizopus* spp., *Aspergillus flavus*, *Aspergillus niger*, *Fusarum* spp., *Chaetomium* spp.	[36]
Chinese ginseng	<1.0 × 10^2^–1.2 × 10^6^	*Bacillus* spp.	<1.0 × 10^2^–6.0 × 10^4^	*Alternaria alternata*, *Aspergillus niger*, *Aspergillus* spp., *Cladosporium* spp., *E. chevalieri*, *Penicillium* spp., *Rhizopus* spp.	[36]
Goji berry (*Lycium barbarum*)	3.5 × 10^2^–7.6 × 10^3^	*Clostridium* spp.	<1.0 × 10^1^–5.0 × 10^2^	-	[37]
Milkvetch root (*Astragalus membranaceus*)	2.0 × 10^2^–9.0 × 10^3^	*Clostridium perfringens*, *Clostridium* spp.	<1.0 × 10^1^–1.0 × 10^2^	-	[37]
Artichoke (*C. scolymus* L.)	1.3 × 10^6^	*Micrococcus* spp., *Staphylococcus* spp.	-	-	[35]
	1.0 × 10^1^–3.0 × 10^5^	*Bacillus* spp., *Micrococcus* spp.	1.0 × 10^1^–2.0 × 10^2^	*Alternaria* spp., *Aspergillus* spp., *Cladosporium* spp.	[2]

TAMC—total aerobic microbial count; TYMC—total yeasts/moulds count.

**Table 2 ijerph-17-06837-t002:** Mycotoxins contaminants in dietary supplements.

Type of Mycotoxin	Toxic Effects and Diseases	Example of Food Supplements	Ref.
Aflatoxin (AF)	carcinogenic, hepatotoxic, immunotoxic, (decreasing immune systems, affecting the structure of DNA, hepatitis, bleeding, kidney lesions)	Liquorice root	[70,71]
		Green tea	[72]
		Ginkgo biloba	[73]
		Milk thistle	[62,74]
[*Aspergillus*]		Ginger	[75,76]
		Ginseng	[77]
		Ginseng root	[47,78]
		Mint	[74,75,79]
		Chamomile flower	[74]
Ochratoxins (OTA, OTB, OTC)	carcinogenic, cepatotoxic, immunotoxic, nephrotoxic, (kidney and liver damage, loss of appetite, nausea, vomiting, suppression of immune system, carcinogenic)	Green coffee	[62]
		Grape	[68]
		Brewer’s yeast	[69]
		Ginger	[76]
[*Aspergillus*, *Penicillium*]			
		Ginseng	[77]
		Mint	[74,79]
		Chamomile flower	[74]
		Liquorice root	[70,71]
Trichothecenes (type A trichothecenes, type B trichothecenes) [*Fusarium*, *Myrothecium*, *Stachybotrys*, *Trichoderma*]	immunotoxic, neurotoxic, (skin necrosis, hemorrhage, anemia, granulocytopenia, oral epithelial lesions, GIS lesions, hematopoietic, alimentary toxic aleukia (ATA), hypotension, coagulopathy)	Ginkgo biloba	[73]
		Different plant	[64]
		Milk thistle	[64,74]
		Mint	[74]
		Chamomile flower	[74]
Zearalenones (ZEN, α-ZOL, β-ZOL, ZAN) [*Fusarium*]	immunotoxic, oestrogenic, teratogenic, (hormonal imbalance estrogenic effect, reproductive problems)	Different plants	[64]
		Ginger	[80]
		Milk thistle	[63,74]
		Mint	[74,79]
		Chamomile flower	[74]
Fumonisins (FB_1_, FB_2_, FB_3_) [*Fusarium*]	carcinogenic, hepatotoxic, immunotoxic, nephrotoxic, neurotoxic (encephalomalacia, pulmonary edema, carcinogenic, neurotoxicity, liver damage, heart failure, esophageal cancer in humans)	Green coffee	[66]
		Milk thistle	[74]
		Mint	[79]
		Chamomile flower	[74]
		Liquorice	[74]
Deoxynivalenol (DON) [*Fusarium*]	inadequate evidence of carcinogenicity	Different plants	[64]
	(interfere with mammalian cellular processes including DNA replication and protein synthesis)	Ginger	[80]
		Milk thistle	[64]
		Mint	[74]
		Chamomile flower	[74]
Citrinin (CIT) [*Aspergillus*, *Penicillium*]	nephrotoxic, reproductive toxicity, teratogenic and embryotoxic effects	Different (plant-based and Red yeast rice)	[81]
		Red yeast rice	[82]
		Mint	[74]
		Chamomile flower	[74]

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
