# Peer review of "Quality of Dietary Supplements Containing Plant-Derived Ingredients Reconsidered by Microbiological Approach"

_ijerph, 2020, doi:10.3390/ijerph17186837_

Round 1

Reviewer 1 Report

General comment:

The paper makes an attempt to review existing data about microbial quality of dietary supplements. The works does not deep into too much into the collected information, and it merely presents values in a very descriptive way, with a limited discussion and elaboration. In some case, information is trivial and too general as in the case of introduction and the first part of the mycotoxin section (4). Other sections are overlapping others and seem to be a bit redundant such as 3.1-3.3. Some information is misleading or not well detailed, e.g. applicable regulation. No clear conclusions seems to be drawn from the data analysis and review. Below some comments if they are useful to improve the work.

Other comments:

The title is not indicative of the paper content. Most of the works analysed are about herbal preparation, and title is referring to a more ample range of products.

In abstract:

"Bad quality" does not mean that it is harmful. Please, be more specific.

Change to “Point to”

Change to  bacterial and fungal contaminations as well as…

Unprecise: “We have showed that the described…”

Chante to “risk that may”

In Introduction:

Introduction is a bit out of focus and  does not introduce the paper topic which is related to microbiology quality and safety of the dietary supplements.

The sentence “They have become an alternative to synthetic drugs” is misleading. Supplement and drugs are/have different things and purposes. Clarify.

OTC is not defined.

In legal aspects:

The below sentences does not say any specific thing about the EU regulation on diet supplements “One of the most important legal acts in the European Union is the regulation (EC) No. 178/2002 of the European Parliament and of the Council of 28 January 2002, defining the general rules and regulations of the food law, establishing the European Food Safety Authority (EFSA), and introducing food safety procedures. The regulation applies to all phases of food manufacturing, processing, and distribution to ensure human health and life [16]. 

Please, refer to the current EU regulation on this type of products.

What does “Inconsistencies” mean in this context? Please, provide the detail on “these inconsistencies”

This statement “Herbal preparations and ……effectiveness or toxicity” is confusing if the herbal preparation are included in dietary supplements. As they are considered foods, they are regulated by the food legislation so they have to comply with all food safety criteria (including microorganisms, toxins, etc) .

In Microbiological Contamination Sources:

Change to “vegetative bacteria” instead of “live bacteria”

The sentence “. It was proven that exposing them to water operation causes the loss of bioactive molecules, and the fungi which grow on the leaves are the main cause of such loss.” is unclear and not well connected to the previous one.

Please, elaborate on the different preventive measures for microbiological contamination, and not just refer to other works, without any detail.

Figure 1 is not needed.

3.2. Quantitative and Qualitative Bacterial Contamination of Dietary Supplements

 This section is just a kind of list of articles and results, but it does not provide a critical analysis of them, and does not provide any conclusion on the main microbial groups, prevalence, etc.

Any comment on the Klebsiella pneumoniae, some suggestion for the presence of E. coli in this study?

which means Bacillus bacteria? Bacteria belonging to the genus Bacillus? If so, these bacteria are spore formers, and there is a special section for them. Any comment of them, why they were majorit.

Concerning the 29 and 150 herbal products, are they considered dietary supplements?

No discussion is provided on why these non-pathogenic and pathogenic bacteria were found in the different samples and studies. Also, why some studies detected pathogens and other not…how often are the pathogenic bacteria in this type of products.

Table 1.  what means in the table “qualitative (CFU/g)” what does it refer to? Total counts? if so, it should be clarified

Mycotoxins

lines 17-69 are  mostly intended to define type of mycotoxins and does not contribute too much on mycotoxins in dietary supplements. It should significantly shortened

Section 5 could be better a justification of the work in introduction

This reviewer could reconsider major revision, but authors should reformulate thoroughly the work, basically providing more discussion to make the paper useful, as a review, and reorganizing and removing some unnecessary sections.

Author Response

General comment:

The paper makes an attempt to review existing data about microbial quality of dietary supplements. The works does not deep into too much into the collected information, and it merely presents values in a very descriptive way, with a limited discussion and elaboration. In some case, information is trivial and too general as in the case of introduction and the first part of the mycotoxin section (4). Other sections are overlapping others and seem to be a bit redundant such as 3.1-3.3. Some information is misleading or not well detailed, e.g. applicable regulation. No clear conclusions seems to be drawn from the data analysis and review. Below some comments if they are useful to improve the work.

Dear Reviewer,

Thank you for your constructive comments and specific suggestions to improve the manuscript content. We appreciate your participation in reviewing our paper.

To increase  the readability of our manuscript, introduction and the first part of the mycotoxin section have been significantly shortened and rewritten. We have rewritten some parts of the paper to provide more clarity and to improve the discussion.

Other comments:

The title is not indicative of the paper content. Most of the works analysed are about herbal preparation, and title is referring to a more ample range of products.

We changed title on: “Quality of Dietary Supplements Containing Plant-Derived Ingredients Reconsidered by Microbiological Approach”

In abstract:

We improved the abstract as suggested:

"Bad quality" does not mean that it is harmful. Please, be more specific.

We changed this sentence (Page 1 line 15)

Change to “Point to”

We changed (Page 1 line 16)

Change to  bacterial and fungal contaminations as well as…

We changed as suggested (Page 1 line 18 )

Unprecise: “We have showed that the described…”

We changed this sentence (Page 1 line 19)

Chante to “risk that may”

We changed as suggested (Page 1 line 23)

In Introduction:

Introduction is a bit out of focus and  does not introduce the paper topic which is related to microbiology quality and safety of the dietary supplements.

We've rewritten the Introduction sections

The sentence “They have become an alternative to synthetic drugs” is misleading. Supplement and drugs are/have different things and purposes. Clarify.

We removed this sentence

OTC is not defined.

We explained the abbreviation – OTC -  ang. Over The Counter Drug (Page 2 line 48)

In legal aspects:

The below sentences does not say any specific thing about the EU regulation on diet supplements “One of the most important legal acts in the European Union is the regulation (EC) No. 178/2002 of the European Parliament and of the Council of 28 January 2002, defining the general rules and regulations of the food law, establishing the European Food Safety Authority (EFSA), and introducing food safety procedures. The regulation applies to all phases of food manufacturing, processing, and distribution to ensure human health and life [16]. 

Please, refer to the current EU regulation on this type of products.

We referred to the current EU regulation.  We changed this paragraph and we added the sentences (Page 4 lines 148-156):

“In European Union food supplements are classified as foodstuffs and thus all food law applies to food supplements. Since 2002, the European Union has created a legal and regulatory framework for these products with the Food Supplements Directive 2002/46/EC. The directive defines a food supplement as “foodstuffs the purpose of which is to supplement the normal diet and which are concentrated sources of nutrients or other substances with a nutritional or physiological effect, alone or in combination, marketed in dose form, namely forms such as capsules, pastilles, tablets, pills and other similar forms, sachets of powder, ampoules of liquids, drop dispensing bottles, and other similar forms of liquids and powders designed to be taken in measured small unit quantities”.

What does “Inconsistencies” mean in this context? Please, provide the detail on “these inconsistencies”

We changed the sentence (Page 4 lines 154-157)

“Based on the analysed data, the authors claim that the frequency of reported cases of inconsistencies in food supplements, as compared with other food products, is high and most of the reported inconsistencies is associated with improper labelling, in particular with falsifying the ingredients and nutritional/health statements.”

to:

“Based on the analysed data, the authors claim that most of the reported inconsistencies is associated with improper labelling, in particular with falsifying the composition, nutrition, and health claims. The frequency of reported cases of inconsistencies in food supplements, as compared with other food products, is high.”

This statement “Herbal preparations and ……effectiveness or toxicity” is confusing if the herbal preparation are included in dietary supplements. As they are considered foods, they are regulated by the food legislation so they have to comply with all food safety criteria (including microorganisms, toxins, etc) .

We changed this sentence

In Microbiological Contamination Sources:

Change to “vegetative bacteria” instead of “live bacteria”

We changed as suggested ( Page 5 line 193)

The sentence “. It was proven that exposing them to water operation causes the loss of bioactive molecules, and the fungi which grow on the leaves are the main cause of such loss.” is unclear and not well connected to the previous one.

We changed this sentence

Please, elaborate on the different preventive measures for microbiological contamination, and not just refer to other works, without any detail.

We elaborated on the different preventive measures for microbiological contamination (Page 5 and 6 lines 237-271)

Figure 1 is not needed.

We deleted Figure 1

3.2. Quantitative and Qualitative Bacterial Contamination of Dietary Supplements

 This section is just a kind of list of articles and results, but it does not provide a critical analysis of them, and does not provide any conclusion on the main microbial groups, prevalence, etc.

We rewrote this section. We combined sections 3.2. “Microbiological quality of dietary supplements containing plant materials” with section 3.3. “Bacteria Spores”.

Any comment on the Klebsiella pneumoniae, some suggestion for the presence of E. coli in this study?

We added: “The presence of bacteria from Enterobacteriaceae family in the tested supplements is mainly associated with human or animal feces used as plant manure.” (Page 8 lines 287-288)

which means Bacillus bacteria? Bacteria belonging to the genus Bacillus? If so, these bacteria are spore formers, and there is a special section for them. Any comment of them, why they were majorit.

We changed “Bacillus bacteria” to “Bacillus spp.”. We rewritten this section and linked it to the previous section. (Page 8)

Concerning the 29 and 150 herbal products, are they considered dietary supplements?

The authors don’t let us know if are these  preparations a dietary supplements.

No discussion is provided on why these non-pathogenic and pathogenic bacteria were found in the different samples and studies. Also, why some studies detected pathogens and other not…how often are the pathogenic bacteria in this type of products.

            The summary of this section has been changed.

Table 1.  what means in the table “qualitative (CFU/g)” what does it refer to? Total counts? if so, it should be clarified

We  changed  “Quantitative (CFU/g)”  to ”Quantitative  TAMC (CFU/g)” and “Quantitative  TYMC (CFU/g)”. The legend explains the abbreviations:

TAMC – total aerobic microbial count

TYMC – total yeasts/moulds count

Mycotoxins

lines 17-69 are  mostly intended to define type of mycotoxins and does not contribute too much on mycotoxins in dietary supplements. It should significantly shortened

We shortened the first part of the mycotoxin section significantly (Page 5 and 6)

Section 5 could be better a justification of the work in introduction

We've rewritten this section and moved to the introduction

This reviewer could reconsider major revision, but authors should reformulate thoroughly the work, basically providing more discussion to make the paper useful, as a review, and reorganizing and removing some unnecessary sections.

Reviewer 2 Report

The authors aimed to analyze microbiological quality and safety of use of dietary supplements containing plant materials. They presented the latest reports referring to bacterial and fungal contamination and the presence of mycotoxins.

The article has been well written. It provides important and interesting informations but some changes are needed. I have noticed the following suggestions and mistakes:

  • Pay attention to the spelling of the bacterial species, write the full name when using it for the first time.
  • Use the term „microbiota“ (instead of microflora) throughout the paper.
  • Avoid the phrase "microbes".
  • Please explain the abbreviation – OTC.
  • Page 2 line 9: delete “better” in this sentence.
  • The names of the microorganisms should be written in italic (in some cases should be corrected if it doesn't result from the technical reasons).

Author Response

The authors aimed to analyze microbiological quality and safety of use of dietary supplements containing plant materials. They presented the latest reports referring to bacterial and fungal contamination and the presence of mycotoxins.

Dear Reviewer,

Thank you for your comments and suggestions. We appreciate your participation in reviewing our paper.

The article has been well written. It provides important and interesting information but some changes are needed. I have noticed the following suggestions and mistakes:

    Pay attention to the spelling of the bacterial species, write the full name when using it for the first time.

We checked the spelling of the bacterial species and we wrote  the full name when used it for the first time.

    Use the term „microbiota“ (instead of microflora) throughout the paper.

We changed the term microflora to microbiota throughout this article (Page 3 line 125 and page 4 line 174)

    Avoid the phrase "microbes".

We changed the term“microbes” to “microorganisms” (Page 2 line 73, Page 3 line 131, Page7 line 219, 220 and 227, Page 16 line 230, Page 17 line 276 and 285)

    Please explain the abbreviation – OTC.

            We explained the abbreviation – OTC -  ang. Over The Counter Drug (Page 2 line 48)

    Page 2 line 9: delete “better” in this sentence.

We removed " better" in this sentence

    The names of the microorganisms should be written in italic (in some cases should be corrected if it doesn't result from the technical reasons).

We checked the spelling of P. aeruginosa. All Latin names were in italics. It seems that the mistakes were due to technical reasons.

Reviewer 3 Report

Section 3.2 paragraph 2, please change ‘Polish chemist’s’ to ‘Polish chemists’.

Page 10, line 57 should be changed to ‘little is known’.

Page 13, line 158: What is the numerical value of this intake? Please specify.

Page 13, line 171: Please clarify ‘the coexisting diseases and the use of different drugs’. Does this mean ‘pre-existing conditions?’

Page 14, section 6: it would be good to state the lower limits of detections of these methods and the levels of mycotoxins detected in some of these studies.

Further comments:

Overall quality of the manuscript is good.

It would be interesting if the authors could comment a bit further on the use of probiotics (as mentioned briefly by the authors on Page 14, lines 219-).

Please briefly comment on the various methods used to tackle the issue of this microbial contamination.

Author Response

Dear Reviewer,

Thank you for your comments and specific suggestions to improve the manuscript content. We appreciate your participation in reviewing our paper.

We have taken into consideration all the suggestions and we've added a detailed description of the changes:

Section 3.2 paragraph 2, please change ‘Polish chemist’s’ to ‘Polish chemists’.

We changed the term ‘Polish chemist’s’ to ‘Polish chemists’

Page 10, line 57 should be changed to ‘little is known’.

We changed the term “little known” to  “little is known”.

Page 13, line 158: What is the numerical value of this intake? Please specify.

Supplemented by adding in parentheses:” TDI is 0,06 µg/kg bw”

Page 13, line 171: Please clarify ‘the coexisting diseases and the use of different drugs’. Does this mean ‘pre-existing conditions?’

We changed “the coexisting diseases and the use of different drugs” to “pre-existing medical conditions and the use of different drugs

Page 14, section 6: it would be good to state the lower limits of detections of these methods and the levels of mycotoxins detected in some of these studies.

We thank the Reviewer for this suggestion. Since the lower limit of detection is one of the crucial parameters of the analytical methods, the following sentences have been added into the manuscript:

 The analytical methods are usually developed to achieve limits of quantification (LOQs) and detection (LODs) in agreement with regulatory authorities for the official control methods [92]. However, these parameters are strongly dependent on the type of used instruments and their actual condition. (Page 19 Lines 258-261)

LOD was different for all detected analytes and dependent on the used matrix. For example, LOD for chanoclavine was 0.04 μg/kg for hazelnuts and 0.1 μg/kg for apple puree. While LOD for patulin was 1.2 μg/kg for hazelnuts and 35.9 μg/kg for apple puree. The proficiency tests materials were performed, and they allowed for example to detect 1406 μg/kg of fumosin B or 7 μg/kg of ochratoxin A in maize. The paper presents detailed information with LODs and LOQs values as well as levels of detected analytes in tests involving different matrices. (Page 19 Lines 263-269)

LODs for analytes identified in the bakery products were below 6 μg/kg and LOQs were in the range 5-100 μg/kg [98]. (Page 19 Lines 278-279)

Further comments:

Overall quality of the manuscript is good.

It would be interesting if the authors could comment a bit further on the use of probiotics (as mentioned briefly by the authors on Page 14, lines 219-).

We added a sentence -  “The composition of probiotic products should include strictly characterized strains of microorganisms of proven clinical effectiveness. In order to reliably confirm the identity of probiotic strains at the species level, tests of such preparations should include not only phenotypic but also genotypic methods.”(Page 3 lines 129-133)

Please briefly comment on the various methods used to tackle the issue of this microbial contamination.

We added a sentence – “So far, methods based on microbial culture and biochemical identification (membrane filtration, plate-count methods, most-probable-number method) are recommended for the quantitative and qualitative evaluation of microorganisms in dietary supplements. However, methods based on molecular biology and omics technologies allow for more sensitive and specific food contaminants identification.“ ( Page 16 lines 231-236)

Round 2

Reviewer 1 Report

The authors addresses most of the comments. Still some minor aspect should be resolved before publication.

Introduction

The starting paragraphes of introduction should be more focused  on plant-based dietary supplements (lines 47-60), and justify them, providing more specific details on this type of supplements since the paper mainly focuses on that. For example, this type of supplements, incorporating plan ingredients, is prone to be more microbiologically risky due to the type of sources, and treatment applied, in comparison to those obtained from bioengineering.

Section 3.2: 

This sentence "Bacillus and Clostridium are deemed to be potentially pathogenic and may cause food poisoning." is not correct. No all bacillus are pathogenic, and there are several which are spoilage in foods and other are saprophites, the same for Clostridium, although there are several species which are patogenic.  It is true that Clostridium can be a microorganism indicator, but the group is not itself pathogenic. Please, makes this clear and correct the statement. 

Author Response

The authors addresses most of the comments. Still some minor aspect should be resolved before publication.

Dear Reviewer,

Thank you for your comments and specific suggestions. We appreciate your participation in reviewing our paper.

Introduction

The starting paragraphes of introduction should be more focused  on plant-based dietary supplements (lines 47-60), and justify them, providing more specific details on this type of supplements since the paper mainly focuses on that. For example, this type of supplements, incorporating plan ingredients, is prone to be more microbiologically risky due to the type of sources, and treatment applied, in comparison to those obtained from bioengineering.

We agree with this comment. We added as suggested (page 2, lines 61-66):

“Dietary supplements incorporating plant ingredients, are prone to be more microbiologically risky due to the their origin, and treatment applied, in comparison to those obtained from bioengineering or organic synthesis. Microbiological contaminants of plant-derived supplements typically include microorganisms which occur naturally in the soil, air and water. Moreover, volatility and sensitivity of the active components of the plants to heat, UV, β and γ irradiation do not permit the use of many decontamination methods.”

Section 3.2:

This sentence "Bacillus and Clostridium are deemed to be potentially pathogenic and may cause food poisoning." is not correct. No all bacillus are pathogenic, and there are several which are spoilage in foods and other are saprophites, the same for Clostridium, although there are several species which are patogenic.  It is true that Clostridium can be a microorganism indicator, but the group is not itself pathogenic. Please, makes this clear and correct the statement.

We modified as suggested and added this sentence (page 9, section 3.2):

“Most species of the genus Bacillus is saprophytic bacteria but Bacillus cereus, often isolated from plants, is associated with foodborne illnesses. Likewise, most bacteria of the genus Clostridium are saprophytes. As commensals, they are part of the physiological intestinal flora of humans and animals. Few species of this genus are human pathogens (e.g. C. botulinum, C. perfringens may cause food poisoning).”

Reviewer 3 Report

The author's response to the comments is satisfactory and I recommend the article for publication.

Author Response

Dear Reviewer,

Thank you. We appreciate your participation in reviewing our paper.